# Hydrogen Production by N-Heterocycle Dehydrogenation over Pd Supported on Aerogel-Prepared Mg-Al Oxides

**Danil M. Shivtsov, Anton P. Koskin \***[ID]**, Sergey A. Stepanenko**[ID]**, Ekaterina V. Ilyina**[ID]**, Artem B. Ayupov, Alexander F. Bedilo**[ID] **and Vadim A. Yakovlev**

Federal Research Center, Boreskov Institute of Catalysis SB RAS, Lavrentyev Av. 5, Novosibirsk 630090, Russia
\* Correspondence: koskin@catalysis.ru

**Abstract:** Tetradecahydrophenazine (14HP) is a nitrogen-containing heterocycle compound with a high content of hydrogen that can be released during its dehydrogenation to phenazine (P). The high stability of the 14HP/P pair and relatively low dehydrogenation temperature make 14HP a promising organic hydrogen carrier. This manuscript is devoted to the investigation of hydrogen production by 14HP dehydrogenation over Pd supported on a series of magnesium-aluminum oxides prepared by the aerogel method. This technique made it possible to synthesize catalyst supports characterized by a high surface area and high concentration of surface active sites where active transition metals could be stabilized in a finely dispersed state. The synthesized aerogels had high specific surface areas and pore volumes. A surface area as high as 600 $m^2/g$ after calcination at 500 °C was observed for the mixed aerogel with an Mg:Al ratio of 1:4. An increase in the concentration of acidic electron-acceptor sites determined by EPR on the surface of the mixed magnesium-aluminum oxide supports with a high surface area prepared by the aerogel method was found to result in higher hydrogen production due to the faster dehydrogenation of sterically hindered nitrogen-containing tetradecahydrophenazine heterocycles.

**Keywords:** hydrogen storage; N-heterocycles; Pd catalysts; acceptorless dehydrogenation; aerogel supports; porous structure; nitrogen adsorption; surface active sites



## 1. Introduction

The widespread use of fossil and other carbon-containing fuels has led to a significant increase in the content of greenhouse gases in the atmosphere [1]. Today, hydrogen, which is an environmentally friendly and renewable energy source, is considered to be a promising alternative fuel. Hydrogen has the highest known gravimetric energy density (143 MJ/kg, which is three times higher than that of gasoline). However, its volumetric energy density is very low (0.0108 MJ/L, which is 3000 times lower than that of gasoline) [2,3]. This is the main aspect hindering the widespread introduction of hydrogen energy. The development of methods for its energy-efficient storage and transportation is an important step in the progress of hydrogen energy implementation [4].

In the last decade, various methods of $H_2$ accumulation and storage have been studied. It has been suggested that hydrogen should be stored in compressed [5], cryo-compressed [6], liquefied [7], and adsorbed states [8], as well as in the form of metal hydrides [9], complex hydrides [10], and organic hydrogen carriers [2,3,11]. According to a number of feasibility studies [12,13], the storage and transportation of hydrogen using organic carriers is currently considered to be one of the most promising technologies.

Since the 2000s, an intensive search for optimal pairs of substrate molecules acting as organic $H_2$ carriers has been carried out, which would allow one to carry out reversible catalytic hydrogenation/dehydrogenation reactions with high selectivity at moderate temperatures [14–16]. In this regard, the optimization of the hydrogen evolution process (the selection of reaction conditions for dehydrogenation, catalytic systems, and organic

substrates) is very important, since this high-temperature endothermic process could be associated with the decomposition of an organic $H_2$ carrier. It has been repeatedly shown that $H_2$-rich nitrogen-containing heterocyclic molecules are characterized by low dehydrogenation temperatures (140–250 °C) in comparison with those of cycloalkanes (300–400 °C). This fact accounts for the decrease in the decomposition rate of the $H_2$-rich substrate. In addition, heterocycles are less volatile, ensuring the production of high-purity hydrogen [17,18]. One of the most promising N-heterocyclic compounds for hydrogen production is tetradecahydrophenazine (perhydrophenazine, 14HP) [19]. Tetradecahydrophenazine is characterized by a high gravimetric hydrogen capacity, good thermal stability, and a high purity of the hydrogenation process, and 14HP precursors can be obtained from the degradation products of lignocellulose [19]. In our recent study, it was shown that the acceptorless dehydrogenation of 14HP made it possible to achieve higher rates of hydrogen evolution compared to decahydroquinoline [20]. In that study, and in the literature on the acceptorless dehydrogenation of N-heterocyclic compounds in general, Pd-containing systems were shown to be the most efficient catalysts [15,21,22]. Using the dehydrogenation of 12H-NEC as an example, it was shown that the catalytic activity of Pd was much higher than that of Pt, Ru, and Rh [23]. This result was confirmed by the study of catalysts containing these metals under identical reaction conditions.

An equally important aspect during the selection of an effective catalyst for the dehydrogenation of N-heterocycles is the choice of a support for the dispersed metal particles. One of the most promising approaches for improving the properties of the supports is the use of the aerogel method for their synthesis [24–27]. The aerogel method was successfully used for the preparation of V-Mg-containing catalysts for the oxidative dehydrogenation of propane. This method made it possible to obtain nanocrystalline samples characterized by a layered structure, high specific surface area, and uniform distribution of vanadium in the magnesium oxide matrix [28,29]. The aerogel-prepared two-component catalyst exhibited a high catalytic activity that exceeded the activity of catalysts previously described in the literature [28]. In addition to their bifunctionality, an important advantage of aerogel supports is their ability to reduce the active component loading. It was shown that the use of an aerogel support for $Pd/Al_2O_3$ and $Rh/Al_2O_3$ made it possible to significantly reduce the loading of noble metals in three-way catalysts due to the uniform distribution of the metals on the surface [30].

Recent studies have shown that the use of $MgAl_2O_4$ as a support improves the properties of dehydrogenation catalysts. For example, $Pd/MgAl_2O_4$ was shown to have high catalytic activity in the dehydrogenation of dodecahydro-N-ethylcarbazole [31]. This activity was explained by the strong interaction between Pd and $MgAl_2O_4$, leading to the stabilization of small particles. It was also noted that the $MgAl_2O_4$ spinel suppressed the sintering of PdO particles during the low-temperature combustion of methane due to the strong interaction of PdO with the support [32,33]. It was shown that the presence of a positive charge on the Pd atom promoted the rapid desorption of dehydrogenation products and made the active sites more accessible [34].

In summary, the development of Pd catalysts supported on mixed $Mg$-$Al$-$O_x$ aerogels for the dehydrogenation of nitrogen-containing heterocycles seems to be promising. In this case, the aerogel structure of the support provides advanced characteristics of a porous structure, whereas the MgO addition contributes to the stabilization of the Pd nanoparticles in the ultrafine state and changes their charge. In the present study, several $Mg$-$Al$-$O_x$ aerogel supports with different compositions were synthesized. The structure of the prepared materials was investigated by various physical methods. The influence of the surface properties of the mixed aerogel supports on the catalytic properties of these Pd-containing systems in the dehydrogenation of a potential hydrogen carrier, tetradecahydrophenazine, was observed.

## 2. Results and Discussion

### 2.1. Characterization of MgAlO$_x$ Aerogel Supports and Pd Catalysts Based on Them

Aerogels are porous materials based on gels, wherein the solvent is substituted by a gas. Typical methods for the synthesis of aerogels are based on a sol–gel process followed by the drying of the gel under supercritical conditions. Conventional drying results in the substantial deformation of the gel structure due to the capillary hydrostatic pressure of the solvent evaporating from the gel pores caused by the surface tension. Under supercritical conditions, the distinction between the liquid and the gas phases vanishes, resulting in the almost complete disappearance of the surface tension. As a result, the use of supercritical conditions during the synthesis of aerogels makes it possible to preserve the original developed structure of the gel almost intact and synthesize materials with high surface areas and pore volumes [35–38].

In the first step, one-component Al$_2$O$_3$ and MgO and mixed Mg-Al-O$_x$ supports with different Mg/Al molar ratios (4:1, 2:1, 1:1, 1:2, and 1:4) were synthesized by the aerogel method. Pd was deposited on the surface of the synthesized supports using a previously reported technique [20], which was adapted taking into account the specific features of the aerogel supports. Absolute ethanol was used as the solvent during the noble metal deposition to avoid the substantial deterioration of the textural characteristics of the aerogels in water. Additionally, reduction in hydrogen at 500 °C was used instead of reduction with sodium borohydride, as it secured the complete removal of chloride ions [39].

The porous structure of the aerogels before and after palladium supporting was studied by nitrogen adsorption–desorption at 77 K (Figure 1 and Table 1). One can see that the porous structure depended on both the chemical composition of the initial gel and the subsequent post-synthetic modification. Indeed, the isotherms of nitrogen adsorption on the MgO and Pd/MgO samples showed virtually no hysteresis loop formed due to capillary condensation. On the samples M1A1, M1A2, Pd/M1A1, and Pd/M1A2, this loop was very narrow (for example, see Figure 1c). Interestingly, on the samples M1A4 and Al$_2$O$_3$, this loop also was very narrow, while after palladium supporting the form of the loop changed and became typical of mesoporous samples with cylindrical pores (see Figure 1e). On the samples M4A1, M2A1, Pd/M4A1, and Pd/M2A1, the hysteresis loop formed due to capillary condensation was large and visible, as expected for mesoporous materials. The observed difference in the appearance of hysteresis loops suggested that the pores in the studied samples had different forms. Slit-like or wedge-like pores should show no hysteresis loop, whereas cylindrical pores usually demonstrate hysteresis loops of different forms [40]. Thus, we used the slit-like pore model for the samples that demonstrated a narrow loop or no hysteresis loop at all. The cylindrical model was used for the samples that showed a hysteresis loop.

The highest surface area and total pore volume among the synthesized aerogels were observed on the sample M1A4 (Table 1). This sample also showed the highest total pore volume among the supports. However, the porous structure of this sample changed after supporting Pd. The initial wedge-like or slit-like geometry became closer to a cylindrical geometry. Palladium deposition changed the porous structure of all the samples. Both the surface area and the total pore volume were reduced, and the maximum pore size distribution shifted to a larger value. The extent of these changes depended on the chemical composition of the aerogel sample. The MgO and Pd/MgO samples showed the most drastic transformation: the surface area decreased from 220 m$^2$/g to 70 m$^2$/g, and the maximum pore size distribution increased from 9.4 nm to 34 nm. The pore structure comprised different wedge-like and slit-like pores (Figure 1b). Most probably, these pores appeared between parallel or almost parallel platelets and crystals in this oxide. As one can see from the cumulative pore volume curves, after Pd deposition involving the dispersion of the precursor materials in liquid ethanol followed by calcination, the smallest pores disappeared due to the cohesion and sintering of the platelets after capillary contraction in solution. Larger pores could be retained after such treatment.

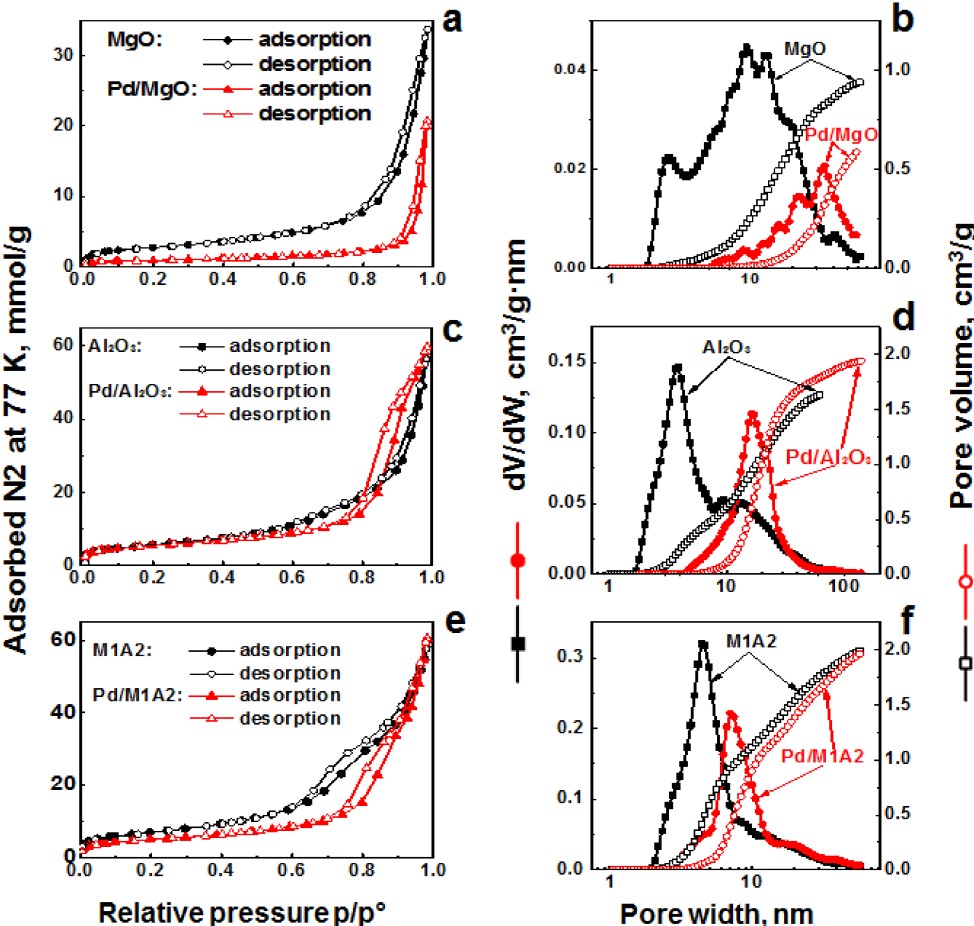

**Figure 1.** Nitrogen adsorption–desorption isotherms (**a,c,e**) and pore size distribution (**b,d,f**) on aerogel-prepared MgO (**a,b**), M1A2 (**c,d**), and Al$_2$O$_3$ (**e,f**) before and after supporting Pd. The differential pore size dV/dW distributions are given by closed symbols, the cumulative volume pore size distributions are given with open symbols. The Broekhoff–de Boer model was used for slit-like pores in MgO, Pd/MgO, M1A2, Pd/M1A2, and Al$_2$O$_3$ and for cylindrical pores in Pd/Al$_2$O$_3$; the regularization factor was 0.1.

Samples M4A1 and M2A1 showed a large hysteresis loop that could be attributed to the fact that the pores in these materials were formed by contact between particles of mixed oxides with large cavities and channels. After Pd deposition, the pore structure was mostly preserved, with the disappearance of the smallest pores to some extent. In the mixed oxides with a higher aluminum content close to that of spinel MgAl$_2$O$_4$, the pores were almost wedge-like or slit-like. The smallest pore size estimated as the maximum of the pore size distribution among the supports and catalysts was obtained for the samples M1A1 and Pd/M1A1, respectively. One could suggest that the cohesion and sintering capability after calcination was lower for this sample than for the others. The increase in the aluminum content made it possible to produce materials with a higher surface area (see samples M1A2 and M1A4, Table 1) and different pore structures, which became mostly cylindrical after supporting Pd (sample M1A4). In agreement with our expectations, the higher aluminum content brought the properties of the sample closer to those of pure alumina, which showed slit-like pores in its pristine form and the cylindrical pores after supporting Pd (sample Al$_2$O$_3$, Table 1).

To determine the phase composition of the synthesized systems, we performed an XRD analysis of the Pd/MgO, Pd/Al$_2$O$_3$, and Pd/M1A2 catalysts. According to the XRD data, only the pure MgO system was well-crystallized (Figure 2, 111, 200, 220). Aerogel-prepared MgO samples calcined at 500 °C are known to consist of MgO nanoparticles with

average dimensions of 4–5 nm [41]. In the mixed oxide, the peaks were shifted to larger angles in comparison with MgO. This meant that the lattice parameter was smaller than in pure MgO, due to the smaller radius of $Al^{3+}$ cations compared to $Mg^{2+}$. Furthermore, the mixed oxide contained cationic vacancies to compensate for the extra charge of the $Al^{3+}$ cations. No peaks attributable to the $MgAl_2O_4$ spinel phase were observed in the XRD pattern of M1A2.

**Table 1.** Characteristics of the porous structure in aerogel-prepared supports and catalysts determined by nitrogen adsorption.

| Sample | SSA, $m^2/g$ | $V_{pore}$, $cm^3/g$ | Pore Size [1], nm | $D_{av}$, nm |
|---|---|---|---|---|
| | | Supports | | |
| MgO | 220 | 1.2 | 9.4 [2] | 21 |
| M4A1 | 415 | 1.6 | 12 [3] | 15 |
| M2A1 | 495 | 1.5 | 8.0 [3] | 12 |
| M1A1 | 535 | 1.4 | 3.3 [2] | 11 |
| M1A2 | 570 | 2.0 | 4.5 [2] | 14 |
| M1A4 | 600 | 2.3 | 5.2 [2] | 15 |
| $Al_2O_3$ | 545 | 2.3 | 3.8 [2] | 17 |
| | | Catalysts 1 wt% Pd | | |
| Pd/MgO | 70 | 0.7 | 34 [2] | 40 |
| Pd/M4A1 | 310 | 1.6 | 16 [3] | 20 |
| Pd/M2A1 | 345 | 1.2 | 9.6 [3] | 14 |
| Pd/M1A1 | 340 | 1.4 | 5.1 [2] | 17 |
| Pd/M1A2 | 410 | 2.1 | 7.1 [2] | 21 |
| Pd/M1A4 | 445 | 1.9 | 14 [3] | 17 |
| Pd/$Al_2O_3$ | 435 | 2.1 | 17 [3] | 19 |

[1] The maximum of the differential pore size distribution dV/dW, where V is the pore volume and W is the pore width for slit-like pores or the pore diameter for cylindrical pores. [2] Slit-like pores were assumed, and the pore size indicates the pore width. [3] Cylindrical pores were assumed, and the pore size indicates the pore diameter.

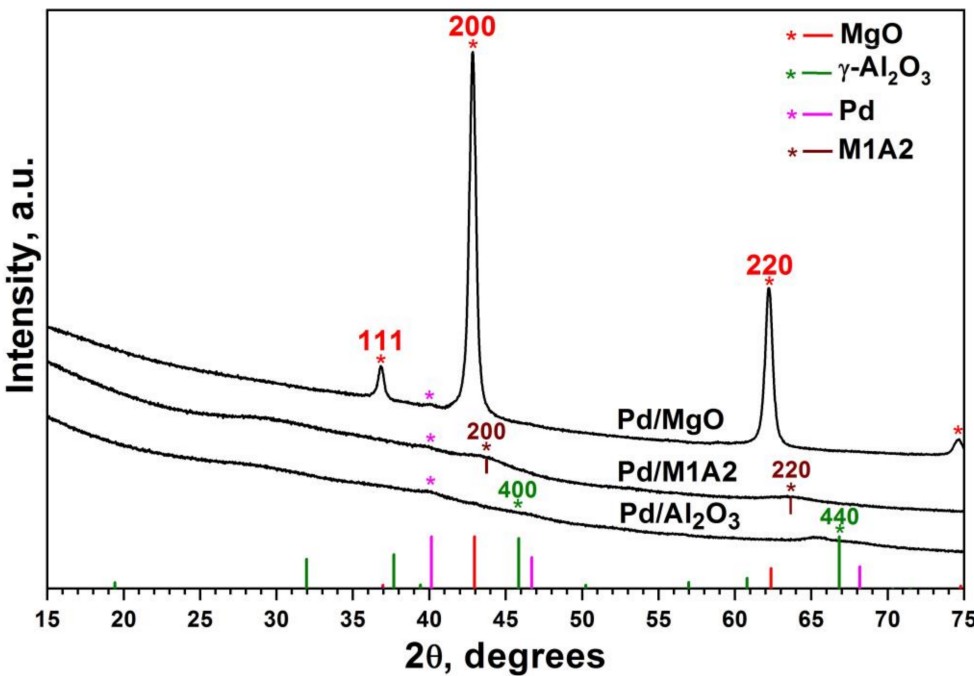

**Figure 2.** XRD diffractograms of MgO, M1A2, and $Al_2O_3$ aerogels.

The Pd/MgO, Pd/$Al_2O_3$, and Pd/M1A2 catalysts were studied by transmission electron microscopy. The TEM micrographs and EDX mapping of the studied catalysts are presented in Figure 3. According to the obtained data, the Pd/$Al_2O_3$ structure consisted of

particles with a flake morphology forming a network. The catalyst-supported magnesium oxide aerogel (Pd/MgO) had a lamellar structure, which was presumably formed as a result of heat treatment during the support preparation step.

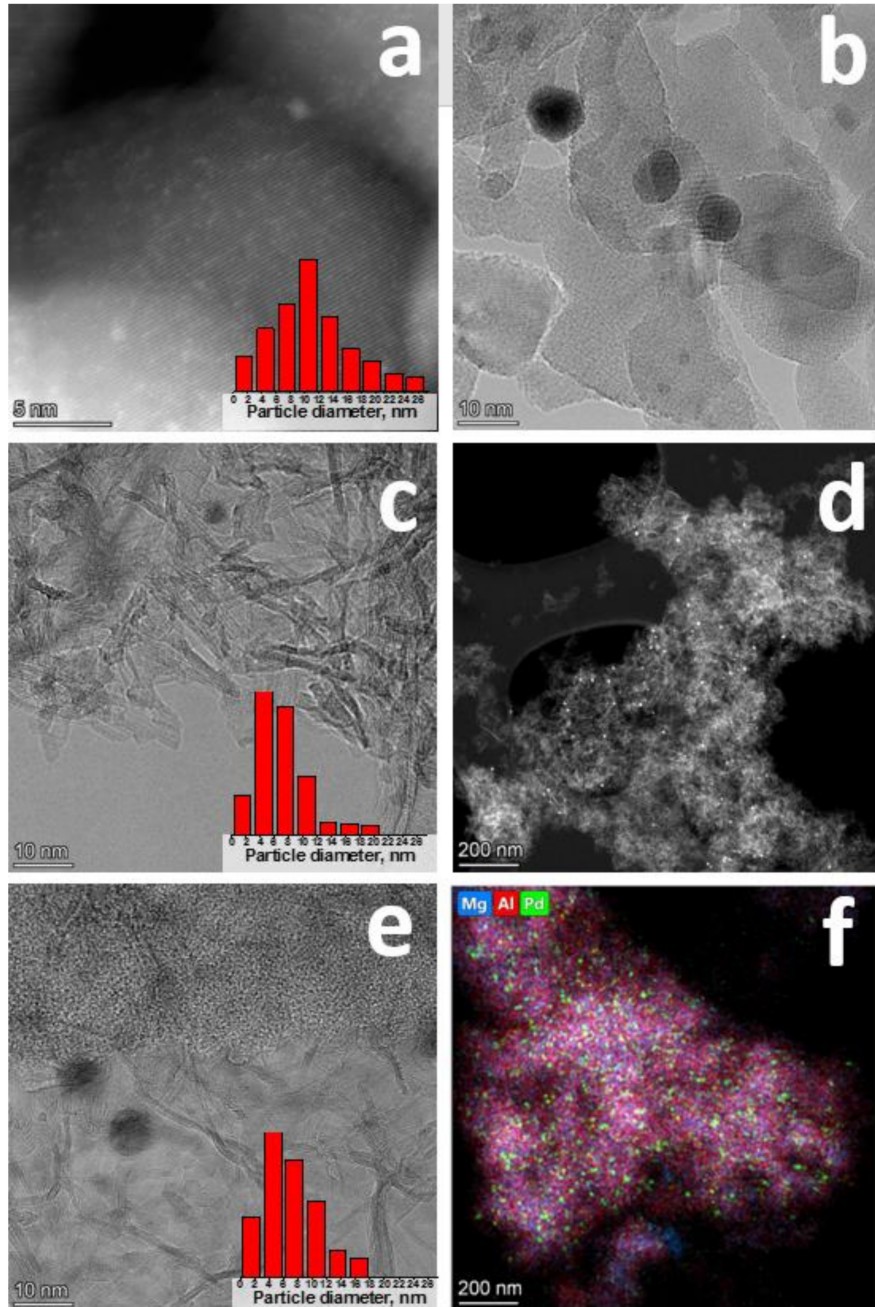

**Figure 3.** HR-TEM images of aerogel catalysts Pd/MgO (**a**,**b**), Pd/Al$_2$O$_3$ (**c**,**d**), Pd/M1A2 (**e**), and EDX of Pd/M1A2 (**f**).

Clusters of palladium nanoparticles about 1 nm in size and particles in the range of 2 to 26 nm with a maximum size of about 10 nm were observed on the MgO surface. The particle size of the palladium on the surface of Al$_2$O$_3$ was in the range of 2–20, with a maximum of about 6 nm. The diameter of the Pd nanoparticles on the mixed oxide M1A2 varied from 2 to 16 nm, with a maximum distribution of about 4 nm. According to the mapping data, Pd was deposited evenly on the surface of the support, whereas Al$_2$O$_3$ and MgO were uniformly mixed with each other.

To determine the possible effect of the support composition on the electronic properties of the supported palladium, the Pd/Al$_2$O$_3$ and Pd/M1A2 catalysts were studied by XPS. Core-level photoelectron peaks corresponding to Pd3$d_{5/2}$ (335.1 eV), O1s (531.3 eV), C1s (284.6 eV), and Al2p (74.5 eV) were observed in the survey spectra of both catalysts. In addition, a peak at 50.9 eV corresponding to Mg2p was observed for Pd/M1A2. The Pd3$d$ core-level spectra of the studied samples are shown in Figure 4. The spectra are described well by the Pd3$d_{5/2}$-Pd3$d_{3/2}$ doublet with a 3:2 intensity ratio. The binding energy of the Pd3$d_{5/2}$ peak was 335.1 eV for both samples. This value is in an excellent agreement with the literature data for palladium in the metal state (335.0–335.2 eV) [42,43]. Thus, it was demonstrated that palladium was in the Pd$^0$ state on the surface of both catalysts, and the presence of basic sites associated with Mg did not have a significant effect on the electronic state of Pd. The ratios of the atomic concentrations of Pd/Al and Pd/(Al+Mg) in the near-surface catalyst layer were similar for both samples and equal to 0.0038 and 0.0035 for Pd/Al$_2$O$_3$ and Pd/M1A2, respectively. These values were close to the theoretical value of 0.005 corresponding to 1 wt.% Pd.

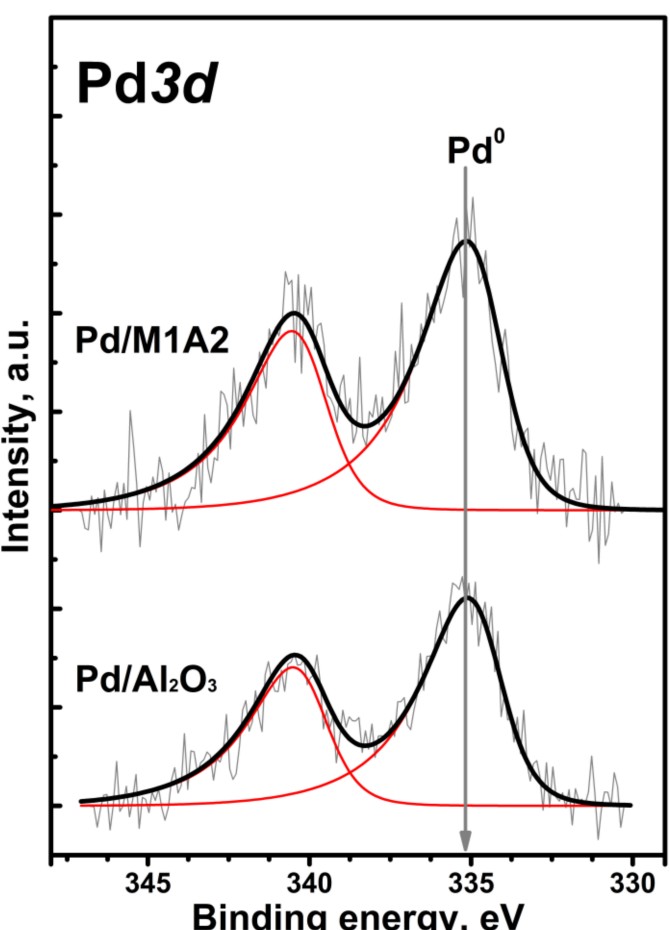

**Figure 4.** Core-level Pd3d XPS spectra of Pd/Al$_2$O$_3$ and Pd/M1A2 catalysts. The spectra were normalized to the integral intensity of the core-level Al2p peaks corresponding to the spectrum of Al$_2$O$_3$.

A series of palladium-containing catalysts were studied by CO pulse chemisorption. This method provided information on the average size of the supported metal species in terms of dispersity determined from CO adsorption data (D$_{CO}$, %; Table 2). According to the obtained results, the presence of Mg in the support favored the preservation of good Pd dispersity after reduction at 500 °C. Presumably, this resulted from the stabilization

of the palladium nanoparticles on basic surface sites of the Mg-Al-O$_x$ supports, in good agreement with the literature data [31–33].

**Table 2.** Dispersity of supported palladium determined by CO chemisorption and concentrations of electron-donor and electron-acceptor sites on the surface of the aerogel-prepared supports.

| Aerogel Supports | Pd Dispersion by CO Chemisorption ($D_{CO}$), % | Active Site Concentration, $10^{18}$ g$^{-1}$ | |
|---|---|---|---|
| | | 1,3,5-Trinitrobenzene (Electron-Donor Sites) | Phenothiazine (Electron-Acceptor Sites) |
| MgO | 46 | 2.4 | 3.2 |
| M4A1 | 50 | 2.4 | 2.4 |
| M2A1 | 54 | 1.8 | 3.7 |
| M1A1 | 54 | 1.1 | 4.4 |
| M1A2 | 55 | 1.2 | 7.5 |
| M1A4 | 49 | 0.7 | 9.6 |
| Al$_2$O$_3$ | 33 | 0.6 | 16.8 |

Electron paramagnetic resonance (EPR) spectroscopy with the use of spin probes makes it possible to study various active sites on the surface of catalysts and sorbents and their role in chemical and catalytic reactions. This method was successfully used for the characterization of electron-acceptor sites, which are capable of abstracting electrons from aromatic probe molecules, and electron-donor sites, which are capable of donating electrons to aromatic nitro compounds [44–49]. As discussed in detail in the cited publications, electron-acceptor sites on metal oxides accounting for the formation of radical cations are associated with strong acid sites, most likely Bronsted. The presence of various metal cations has little or no positive effect. Meanwhile, the electron-donor sites responsible for the opposite process seem to be related to basic sites.

Both probe molecules had one nitrogen atom. Thus, the EPR spectra of the ion radicals formed after their adsorption on the active sites were characterized as three-component spectra with hyperfine splitting on the nitrogen atom (Figure 5). The spectra observed after the adsorption of 1,3,5-trinitrobenzene (TNB) on the aerogel supports (Figure 5a) were similar to the previously reported spectra of TNB radical anions on the Al$_2$O$_3$ surface [42]. They were characterized by a triplet EPR signal with frozen rotation with g = 2.005 and a hyperfine splitting constant A$_{ZZ}$ of about 28 G, which did not depend much on the Mg:Al ratio. This result indicates that electron-donor sites donating a single electron to the TNB molecule, similar to those present on the Al$_2$O$_3$ surface, existed on the surface of all studied aerogel samples. Previously, such sites were shown to stabilize palladium deposited on the Al$_2$O$_3$ surface in a finely dispersed state with high CO oxidation activity [50]. The concentration of strong electron-donor sites revealed immediately after the probe adsorption generally grew with the increase in the Mg concentration in the sample (Table 2). This was quite natural because the increase in the Mg:Al ratio was expected to lead to an increase in the sample basicity. The highest concentration (2.4 × $10^{18}$ g$^{-1}$) was observed on M4A1 and on pure MgO, which had a substantially lower surface area. The concentration of such sites on the Al$_2$O$_3$ aerogel was approximately four times lower.

The ionization potential of phenothiazine is 6.8 eV, making it a good probe for the investigation of weak electron-acceptor sites capable of abstracting an electron from phenothiazine and stabilizing its radical cation on the surface [48,49,51]. A triplet EPR signal with hyperfine splitting on the nitrogen atom A$_{ZZ}$ of about 18 G was observed after phenothiazine adsorption on all the studied samples. This corresponded to adsorbed phenothiazine radical cations. The concentration of such acidic sites decreased with an increase in the Mg:Al ratio, which made the sample more basic and less acidic. The highest concentration of electron-acceptor sites was observed on the Al$_2$O$_3$ aerogel: 16.8 × $10^{18}$ g$^{-1}$ (Table 2). The M4A1 sample was characterized by the lowest concentration of such sites 2.4 × $10^{18}$ g$^{-1}$, which was even lower than on the MgO surface.

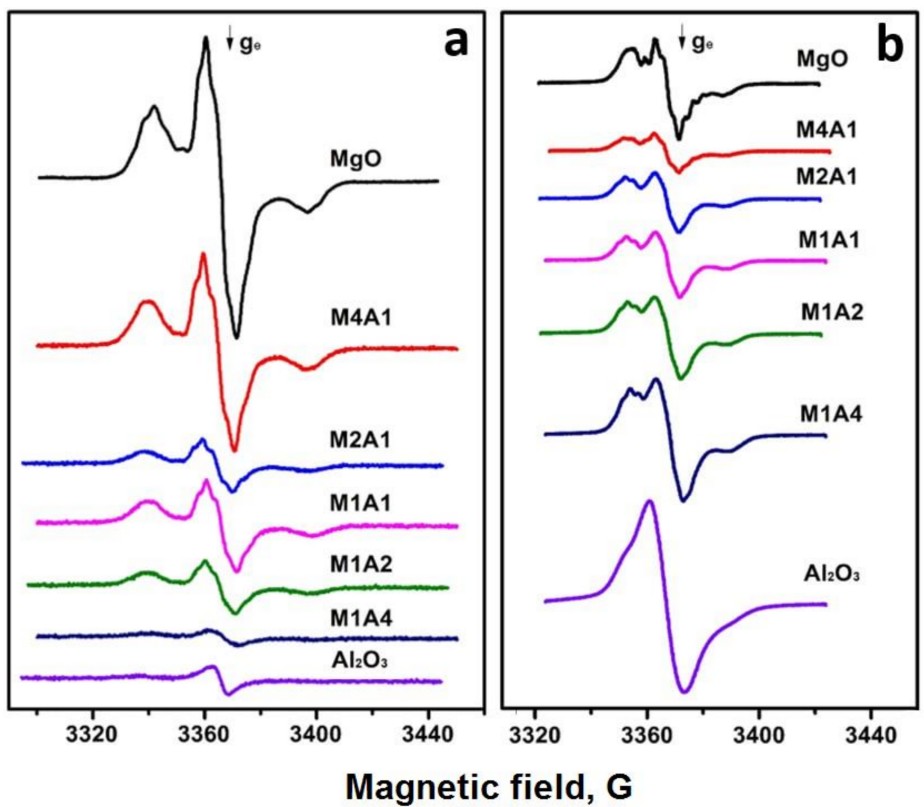

**Figure 5.** EPR spectra observed after adsorption on 1,3,5-trinitrobenzene (**a**) and phenothiazine (**b**) on the studied aerogel supports.

Thus, electron-donor and electron-acceptor sites were present on the surface of all the studied aerogel supports. The observed trends were in a good agreement with the expected increase in the material basicity with the growth of the Mg:Al ratio. Its growth led to an increase in the concentration of basic electron-donor sites and a decrease in the concentration of acidic electron-acceptor sites.

Overall, it was shown that the mixed oxide samples had a high surface area, a high concentration of electron-donor sites where Pd could be stabilized in a finely dispersed active state, and a low concentration of electron-acceptor sites that might lead to unwanted transformations of organic molecules initiated by their acid properties. In the next stage, the catalytic activity of the synthesized materials was studied to determine the optimal Mg:Al ratio.

## 2.2. Investigation of the Catalytic Activity of Aerogel-Supported Pd Catalysts

The catalytic activity of Pd/Mg-Al-$O_x$ catalysts was studied in the dehydrogenation of tetradecahydrophenazine (14HP) using tetraglyme as the solvent. A solvent had to be used because both the reagent 14HP and the reaction product phenazine (P) are solid at the dehydrogenation temperature of 240 °C. In our previous publication [20], the effect of the solvent on 14HP dehydrogenation was studied for 1 wt% Pd/$\gamma$-$Al_2O_3$ catalyst. Polar tetraglyme with a high boiling temperature was shown to be the most preferable solvent. The main parameter characterizing the catalytic activity was the hydrogen yield relative to the maximum theoretically possible hydrogen release ($YH_2$, %). This characterizes the depth of 14HP conversion to dehydrogenation products. In addition, the concentrations of side products 5-methyl-5,10-dihydrophenazine and 5,10-dimethyl-5,10- dihydrophenazine were determined using gas chromatography ($Y_{MP}$, %). The formation of these compounds is related to partial tetraglyme decomposition over the surface acid sites of the support followed by the methylation of the nitrogen atom in phenazine.

The analysis of the reaction mixture probes taken at intermediate dehydrogenation times (10, 20, and 30 min) revealed significant differences in the reaction routes of dehydrogenation between the catalysts containing MgO and the catalysts based on $Al_2O_3$. Typical chromatograms of the reaction mixture probes taken after 30 min of dehydrogenation over Pd/M1A2 and Pd/$Al_2O_3$ are shown in Figure 6. 1,2,3,4,4a,5,10,10a-octahydrophenazine (8HPc) was registered as an intermediate over Pd/M1A2, whereas the reaction mixture composition over Pd/$Al_2O_3$ was similar to that observed previously over Pd/Pural-550 prepared using a commercial $Al_2O_3$ support [20].

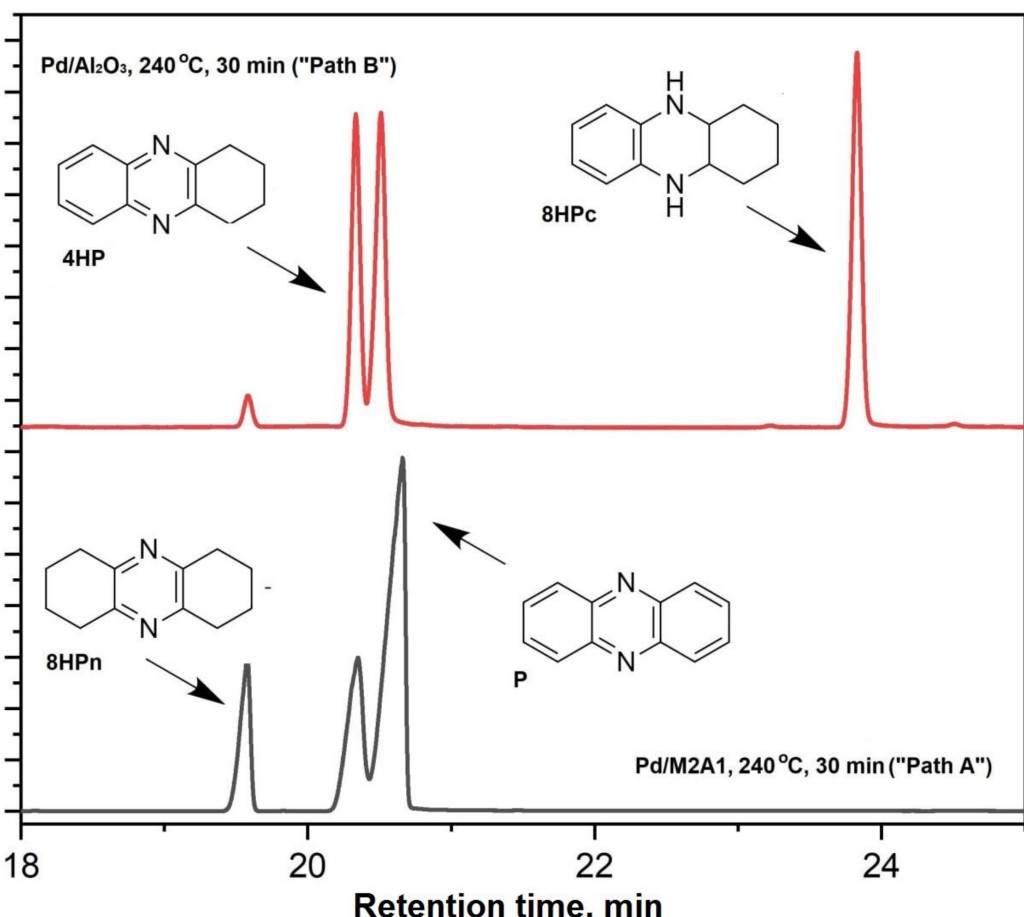

**Figure 6.** Typical chromatograms of the reaction mixture probes taken after 30 min of 14HP dehydrogenation over Pd/M1A2 and Pd/$Al_2O_3$.

As was shown previously [20], hydrogen release over Pd/$Al_2O_3$ begins from the side ring, despite the fact that the dehydrogenation of the central ring in 14HP is more thermodynamically favorable. This reaction route is apparently related to sterical problems for hydrogen release from the central ring in 14HP. For catalysts containing MgO, the hydrogen release began from the dehydrogenation of the central ring (Figure 7). As was shown above, the electronic state of Pd was the same in Pd/$Al_2O_3$ and Pd/M1A2 and corresponded to palladium metal (Figure 4). The average Pd particle size was also very similar (Figure 3). Thus, the presence of basic sites in the supports containing Mg affected the reaction pathway. Apparently, basic sites related to Mg can relieve some of the sterical hindrance, opening up a more thermodynamically favorable pathway [20].

**Figure 7.** Differences in reaction routes of 14HP dehydrogenation over acidic and basic catalysts (**a**), and suggested reaction mechanisms over Pd/Al$_2$O$_3$ (**b**) and Pd/Mg-Al-O$_x$ (**c**).

These observations were in a good agreement with previously obtained results showing that the basic sites of the support in addition to the noble metal play an important role in the hydrogenation and dehydrogenation of nitrogen-containing heterocycles [52–56]. When MgO was used as the support [54], the presence of strong basic sites (surface oxygen atoms) altered the mechanism of quinoline hydrogenation.

Based on the GC data and the previously reported results [50–54], the following reaction mechanisms were suggested for the first stage of 14HP dehydrogenation (Figure 7b,c). According to the proposed mechanisms, dehydrogenation over Pd/Al$_2$O$_3$ occurs in one stage via the abstraction of two hydrogen atoms by a Pd nanoparticle (Figure 7b). In this case, both hydrogen atoms must be oriented in roughly the same direction towards the Pd nanoparticle, which is apparently possible only for the side ring due to the sterical problems. Over supports containing Mg, strong basic sites can participate in the hydrogen abstraction (Figure 7c). In this case, one hydrogen atom is initially abstracted by a basic site, whereas the second atom is taken by a Pd nanoparticle in the following step. Thus, the sterical problems are removed, and the dehydrogenation process can follow the most energetically favorable pathway. In both cases, the eventual coupling of the hydrogen atoms yields molecular hydrogen.

Note that the reaction pathway "Path A" observed over mixed oxide supports is preferable, because 5,10-dihydrophenazine (2HP) is one of intermediates in "Path B"; 2HP is more stable in the desired dehydrogenation reaction and can be accumulated in the

reaction mixture. Furthermore, it is easily methylated to form methylphenazines (MP) as side products.

Figure 8 illustrates the effect of the support on the catalytic properties. Catalysts containing 1 wt% Pd deposited on aerogel-prepared MgO; $Al_2O_3$ oxide supports; mixed oxides (M4A1, M2A1, M1A1, M1A2, and M1A4); and a $\gamma$-$Al_2O_3$ support from commercial Pural SB were studied under identical conditions. Hydrogen was not released when the dehydrogenation experiment was carried out in the absence of the catalyst or when the reaction was performed using pristine supports without Pd. Despite the fact that slightly poorer Pd dispersion was observed according to the CO chemisorption data over the aerogel supports ($D_{CO}$ = 33–55%) in comparison with 1 wt% Pd/$\gamma$-$Al_2O_3$ ($D_{CO}$ = 59%), their activity in 14HP dehydrogenation was significantly higher (Figure 8). The highest hydrogen production yield ($YH_2$ 83%) was observed for the catalysts using the mixed aerogel M1A2 as the support.

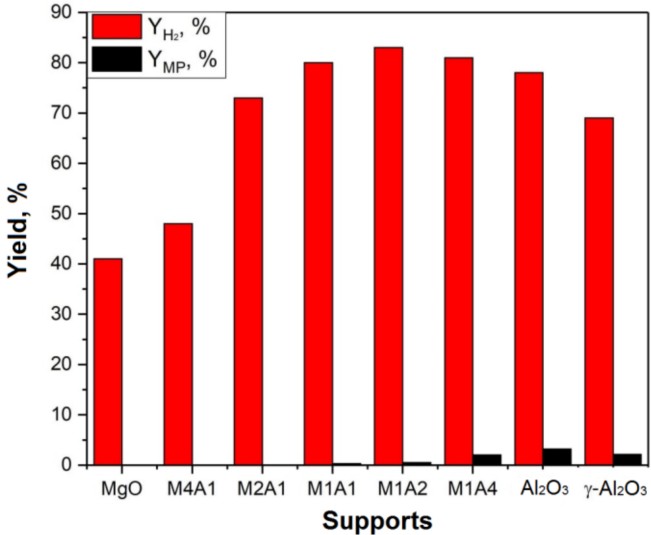

**Figure 8.** Effect of the support on the catalytic properties of catalysts containing 1 wt% Pd in 14HP dehydrogenation. $Y_{H2}$ is the hydrogen yield, % of the theoretical maximum; $Y_{MP}$ is the yield of methylphenazines, %.

The observed dependence of $YH_2$ on the composition of the support was presumably determined by a number of factors. In addition to a high surface area, the most efficient supports (M1A1, M1A, and M1A4), which contained magnesium, were characterized by significant concentrations of basic electron-donor sites ($0.7$–$1.2 \cdot 10^{18}$ $g^{-1}$, Table 2). These sites accounted for the efficient stabilization of the palladium nanoparticles. Additionally, their presence altered the mechanism of 14HP dehydrogenation, favoring the formation of 8HPn. Overall, this led to the higher efficiency of the dehydrogenation process and accelerated hydrogen production. Based on the catalytic activity data, Pd/M1A2 was characterized by an optimal combination of the surface area and concentration of basic sites. According to the XRF data, the Pd concentration in the Pd/M1A2 sample after the catalytic reaction was similar to that observed for the fresh catalyst. The recycled catalyst was studied again in the catalytic reaction. Its catalytic activity was preserved after recycling, remaining within the experimental error limits.

Meanwhile, the concentration of side products resulting from methylation ($Y_{MP}$) also substantially depended on the support used. An increase in the concentration of acidic electron-acceptor sites (Table 2) as well a surface area increase intensified the solvent decomposition reaction, resulting in an increased concentration of the methylation side products in the reaction mixture. Thus, the 1 wt% Pd/M2A1, Pd/M1A1, and Pd/M1A2 catalysts produced using aerogel-prepared mixed oxide supports were the most promising for 14HP dehydrogenation in the presence of tetraglyme, as they demonstrated both good

tetraglyme stability and a high hydrogen production rate. The hydrogen evolution rate observed in this study was compared with the literature data (Table 3).

**Table 3.** Comparison of the hydrogen evolution rate observed in this study with the literature data.

| Catalyst | Reactant | Reaction Conditions | H$_2$ Evolution Rate (mmol g$_{Met}^{-1}$ min$^{-1}$) | Ref. |
|---|---|---|---|---|
| Pd$_2$Ru@SiCN | 14HP | 2 mmol 14HP, 70 mg Pd$_2$Ru@SiCN (0.36 mol. % active metal), 190 °C, 0.75 mL diglyme, 24 h. | 39.31 | [19] |
| Ir complexes (Homog.) | 2,6-dimethyldecahydro-1,5-naphthyridine | 0.25 mmol of reactant and the catalyst under reflux in p-xylene for 20 h, 138 °C, 5 mol% of Ir. | 0.43 | [57] |
| 1 wt% Pd/γ-Al$_2$O$_3$ | decahydroquinoline | 10.83 mmol of reactant, 230 °C, 4 h, M/R = 0.1 mol%. | 62 | [58] |
| 1 wt% Pd/C | | 7.732 mmol of H12-MBP, M/R = 0.1 mol%, 270 °C, 4 h. | 237 | [59] |
| Pd/3.3CCA | 2-[(n-methylcyclohexyl)-methyl]piperidine (MBP) | 7.732 mmol of H12-MBP, M/R = 0.1 mol%, 250 °C, 4 h. | 168 | [60] |
| 3PdA | | 7.732 mmol of H12-MBP, 270 °C, 4 h, M/R = 0.59 mol%. | 27.5 | [61] |
| MPdA600_5h | | 7.3 mmol of reactant, M/R ratio of 0.1 mol%, and 250 °C for 4 h. | 16.69 | [62] |
| 1 wt% Pd/Mg-Al-Ox | 14HP | 1 mmol 14HP in tetraglyme (3 mL), 240 °C, 45 min, 50 mg of catalyst. | 252 | This study |

In the literature, there are very few examples of the catalytic dehydrogenation of perhydro-N-heterocycles consisting of six-membered rings. Meanwhile, this type of N-containing heterocycle is characterized by a high content of hydrogen that could be released. In addition, nitrogen-containing six-membered rings have higher stability with respect to ring opening due to hydrodenitrogenation compared to five-membered rings [63,64].

### 3. Materials and Methods

*3.1. Catalyst Preparation*

Aerogel-prepared magnesium oxide was synthesized according to a procedure adapted from the method pioneered by Klabunde et al. [65]. First, magnesium metal ribbon (1.2 g) was subjected to a reaction with methanol (75 mL). Then, 150 mL toluene was added to the reaction medium. The formed magnesium methoxide was hydrolyzed with a stoichiometric amount of distilled water (1.8 mL). The solvents (methanol and toluene) were purchased from J.T. Baker (Phillipsburg, NJ, USA) and Baza1R JSC (Staraya Kupavna, Russia), respectively. All the materials were used as received.

For the synthesis of aluminum oxide according to the earlier developed method [66], 6 g of aluminum isopropoxide (Alfa Aesar, Ward Hill, MA, USA) was dissolved in a mixture of 75 mL methanol and 150 mL toluene. Next, hydrolysis was carried out with 1.6 mL of distilled water.

For the synthesis of mixed oxide aerogels, first, magnesium metal ribbon was subjected to a reaction with methanol excess (75 mL). Then, aluminum isopropoxide and 150 mL

toluene were added to the reaction mixture. The masses of magnesium metal and aluminum isopropoxide were chosen so that their molar ratios were 1:4; 1:2; 1:1; 2:1; and 4:1. After the mixture was stirred for 1 h, it was subjected to hydrolysis with a stoichiometric ratio of distilled water.

After the addition of water for hydrolysis, each resulting gel was stirred for 16 h. Then, it was placed in a 500 mL autoclave (Amar, Mumbai, India), in which it was heated for 3 h to 270 °C. The final pressure in the autoclave was 80–95 atm. Then, the gas from the autoclave was slowly released at 270 °C, and the autoclave was purged with argon for 5 min. Each synthesized aerogel sample was additionally calcined in air at 500 °C for 3 h at a heating rate of 100 °C/h. The synthesized samples of the mixed oxides were designated as M$XAY$, where $X$ and $Y$ are the amounts of Mg and Al corresponding to their molar ratio.

For the synthesis of 1 wt% Pd/Mg-Al-O$_x$ catalysts, the adsorption precipitation technique was used with H$_2$[PdCl$_4$] as the precursor of the active component. Applying this method, 1 g of the support was dispersed in 20 mL of an EDTA solution (56 mg) in ethanol using a magnetic stirrer (600 rpm). Then, liquid H$_2$[PdCl$_4$] precursor at the required concentration was added dropwise and stirred for one hour (600 rpm). The resulting samples were filtered, washed with ethanol, dried at 120 °C, and calcined at 500 °C. Then, the obtained catalysts were reduced in the hydrogen flow at 500 °C for 1 h with a H$_2$ flow rate of 100 mL/min.

In addition, 1 wt% Pd/γ-Al$_2$O$_3$ catalyst prepared using the commercial precursor Pural SB-1 (Sasol, Hamburg, Germany) according to a previously reported procedure [20] was used for comparison.

*3.2. Catalyst Characterization*

X-ray diffraction patterns of the samples were recorded in the range 2θ 15–75° with a step of 0.05° and an accumulation time of 3 s using a Bruker D8 diffractometer with CuKα radiation (λ = 1.5418 A) (Bruker AXS GmbH, Karlsruhe, Germany). The dispersion of metal particles and the microstructure of the catalysts were studied by transmission electron microscopy. The images were acquired with a JEM-2010 (JEOL Ltd., Tokyo, Japan) operating at an accelerating voltage of 200 kV and a resolution of 0.14 nm. The device was equipped with an energy dispersive X-ray (EDX) spectrometer XFlash (Bruker AXS GmbH, Karlsruhe, Germany) with an energy resolution of 130 eV.

The Pd concentration in the fresh and exhaust catalysts was determined using an ARL Advant'X 2247 X-ray fluorescence spectrometer (Thermo Fisher Scientific Inc., Waltham, MA, USA).

The porous structure was studied by nitrogen adsorption–desorption at 77 K. The isotherms were measured by means of an automatic adsorption analyzer ASAP-2400 (Micromeritics Corp., Norcross, GA, USA). Before the measurements, all samples were degassed in a vacuum of less than 1 Pa at 150 °C for 16 h. All calculations were performed using ASAP 2020 Plus ver. 2.00 software. The desorption branch of the isotherms was used for the calculation of pore size distributions.

The XPS measurements were carried out using a photoelectron spectrometer (SPECS Surface Nano Analysis GmbH, Germany) equipped with a PHOIBOS-150 hemispherical electron energy analyzer and an XR-50 X-ray source with a double Al/Mg anode. The core-level spectra were obtained using AlKα radiation (hν = 1486.6 eV) under ultrahigh vacuum conditions. The binding energy (Eb) of the photoemission peaks was corrected to the Al2p peak (Eb = 74.5 eV) of alumina [67,68]. The curve fitting was carried out using CasaXPS software [69].

Active sites on the surface of the obtained materials were studied by electron paramagnetic resonance using 1,3,5-trinitrobenzene (TNB) and phenothiazine as spin probes. Before the adsorption of the spin probes, the samples were activated by heating in air at 500 °C for 3 h. The probes were adsorbed from their $2 \times 10^{-2}$ M solutions in toluene. EPR spectra were recorded at room temperature with an ERS-221 spectrometer (Center of Scientific Instruments Engineering, Leipzig, Germany).

CO chemisorption was measured in a pulsed mode using a Chemosorb analyzer (MBE, Novosibirsk, Russia) equipped with a thermal conductivity detector. The catalyst (50 mg) was loaded into a U-shaped quartz reactor and treated with an $H_2$ flow (100 mL/min) at 20 and 100 °C with a heating rate of 10 °C/min. The reactor was kept at the final temperature for 20 min, then purged with argon and cooled to room temperature. After cooling, CO pulses (0.1 $cm^3$) were injected into the reactor until the sample was saturated, and the amount of chemisorbed CO was estimated.

### 3.3. Tetradecahydrophenzine Dehydrogenation

Tetradecahydrophenazine was synthesized from a commercially available phenazine substrate (Sigma Aldrich, St. Louis, MO, USA) according to the previously described method [20]. The 1H-NMR and GC-MS methods were used to confirm the purity of the obtained heterocyclic substrate (higher than 99.7%).

To study tetradecahydrophenazine dehydrogenation, a reaction setup similar to that described previously [20] was used. In a standard test of the catalytic activity, the catalyst (50 mg) and a suspension of 1 mmol tetradecahydrophenazine in tetraglyme (3 mL) (Acros Organics, Geel, Belgium) were loaded into the reactor. The system was purged with argon (20 mL/min, 5 min, room temperature), and the reactor was sealed and connected to a weight analysis line. Next, the reaction mixture was heated to 240 °C with stirring (500 rpm) for 15 min (0–900 s), and the dehydrogenation reaction was carried out for 45 min (900–3600 s) before the reactor was removed from the heating zone. The error of the weighing system (i.e., comparison with data on the volume of released gas) was estimated as $\pm 2$ mL $H_2$. When tetraglyme (3 mL) was used as a solvent, the increase in the gas volume in the reactor due to thermal expansion (up to 240 °C) was estimated as $19 \pm 1$ mL. Reaction mixture probes were taken at the end of the reaction and at intermediate dehydrogenation times (10, 20, and 30 min). Then, the selected aliquot of the reaction mixture was separated from the catalyst particles by centrifugation and analyzed by the GC method.

The GC analysis was performed using an Agilent-7890A gas chromatograph (Santa Clara, CA, USA) equipped with a ZB-5HT column. A catalytic experiment using a recycled catalyst was performed using the same procedure after washing the catalyst from the first experiment with acetone and drying in the argon flow at 50 °C for 1 h.

The hydrogen yield characterizing the depth of dehydrogenation was calculated as follows:

$$Y_{H_2} = Y_P + \frac{6Y_{2HP}}{7} + \frac{5Y_{4HP}}{7} + \frac{3Y_{8HPc}}{7} + \frac{3Y_{8HPn}}{7}$$

where Y is the molar fractions of the product ($Y_P$) and intermediates ($Y_{2HP}$, $Y_{4HP}$, $Y_{8HPn}$, $Y_{8HPc}$) obtained by gas chromatography. The dehydrogenation selectivity was estimated from the total yield of methylphenazines ($Y_{MP}$, %), i.e., the sum of the yields of 5-methyl-5,10-dihydrophenazine and 5,10-dimethyl-5,10-dihydrophenazine.

## 4. Conclusions

Mixed magnesium-aluminum oxide supports with different Mg:Al ratios (Mg:Al = 4:1, 2:1, 1:1, 1:2, 1:4) as well as pristine MgO and $Al_2O_3$ were synthesized by the aerogel method with high-temperature autoclave drying. The structure and properties of the synthesized aerogel samples were studied by a number of physical methods (nitrogen adsorption, XRD, HRTEM, EPR, and CO chemisorption). All the synthesized aerogels were characterized by high specific surface areas and pore volumes. Surface areas as high as 600 $m^2$/g after calcination at 500 °C were observed for M1A4. Electron-donor and electron-acceptor sites were observed on the surface of all the studied aerogel supports. An increase in the Mg concentration in the aerogels led to an increase in the concentration of basic electron-donor sites and a decrease in the concentration of acidic electron-acceptor sites. Catalysts containing 1 wt% Pd were prepared by palladium deposition from ethanol solution on the synthesized aerogel supports. The presence of surface active sites provided for Pd stabilization in a finely dispersed state with a nanoparticle size of about 5 nm and a narrow particle size

distribution. It was demonstrated that the high dispersity of the supported metal and the high surface area of the supports had a favorable effect on the catalytic activity in the acceptorless dehydrogenation of tetradecahydrophenazine, leading to a high reaction rate. The promotion of the basic properties by the addition of magnesium to alumina was shown to change the tetradecahydrophenazine dehydrogenation mechanism, making it possible to start the process from the central ring of the heterocycle. The application of tetradecahydrophenazine as the hydrogen source required the use of a solvent such as tetraglyme. The use of catalysts with a low concentration of acidic electron-acceptor sites made it possible to suppress tetraglyme decomposition to form side products (methylphenazines). Overall, 1wt% Pd/M2A1, Pd/M1A1, Pd/M1A2 catalysts prepared with mixed oxide aerogel supports were found to be the most efficient catalysts for the acceptorless dehydrogenation of tetradecahydrophenazine.

**Author Contributions:** Conceptualization, A.P.K.; data curation, A.P.K. and A.F.B.; formal analysis, A.P.K., S.A.S., D.M.S. and E.V.I.; funding acquisition, A.F.B.; investigation, D.M.S., A.P.K., S.A.S., E.V.I. and A.B.A.; methodology, D.M.S., A.P.K., E.V.I. and A.B.A.; project administration, A.P.K. and A.F.B.; resources, V.A.Y. and A.F.B.; supervision, A.P.K. and A.F.B.; validation, A.P.K. and A.F.B.; visualization, D.M.S., S.A.S., E.V.I. and A.B.A.; writing—original draft, A.P.K. and D.M.S.; writing—review and editing, A.P.K. and A.F.B. All authors have read and agreed to the published version of the manuscript.

**Funding:** This research was funded by the Ministry of Science and Higher Education of the Russian Federation within the governmental order for the Boreskov Institute of Catalysis (projects AAAA-A21-121011390007-7 and AAAA-A21-121011390054-1).

**Data Availability Statement:** All data included in this study are available upon request by contacting the corresponding author.

**Acknowledgments:** The characterization of the samples was performed using the equipment of the Center of Collective Use "National Center of Catalysts Research". The authors are grateful to Dmitriy Y. Yermakov for CO chemisorption studies, Svetlana A. Cherepanova for XRD studies, Ekaterina I. Shuvarakova for assistance with EPR studies, Andrey A. Saraev for XPS studies and Evgeniy Y. Gerasimov for the HRTEM studies.

**Conflicts of Interest:** The authors declare no conflict of interest.

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
