# Peer review of "Hydrogen Production by N-Heterocycle Dehydrogenation over Pd Supported on Aerogel-Prepared Mg-Al Oxides"

_catalysts, doi:10.3390/catal13020334_

Round 1

Reviewer 1 Report

This manuscript fabricated a series of Pd catalysts supported on mixed Mg-Al-Ox aero-gels for dehydrogenation of nitrogen-containing heterocycles. The authors used XRD, BET, TEM, EPR and gas chromatography to examine the physical and chemical properties of the Pd/Mg-Al-Ox catalysts. Besides, the authors investigated differences of mechanisms of 14HP dehydrogenation over acid and basic catalysts. I recommend this manuscript for publication in Catalysts after the following points are addressed:

1.     To further illustrate the electronic structure of catalysts, XPS characterizations should be considered.

2.     The authors should provide the hydrogen yield of samples without Pd as a match group in order to indicate the effect on Pd in 14HP dehydrogenation.

3.     The authors point that Pd/M1A4 shows the highest surface area in BET, while the hydrogen yield of Pd/M1A4 is lower than that of Pd/M1A2. The authors should explain the reason.

4.     The stability of a catalyst is an important characteristic. The structure after test should be therefore characterized, such as phase, morphology and chemical states.

5.     The authors should prepare a Table where other relevant works should be included.

Author Response

The authors of the manuscript express their gratitude to the reviewer for a careful reading of the text. Reply to Reviewer 1 submitted as a doc file

Reviewer 2 Report

This paper describes the dehydrogenation of tetradecahydrophenazine over MgAlOx aerogel supported Pd catalysts. The catalytic activity of Pd altered by the Mg/Al ratio and the Pd/M1A2 (Mg/Al=1/2) showed the highest activity. As the authors stated, it has been already reported that Pd/MgAl2O4 was a good catalyst for dehydrogenation of N-heterocyclic compounds. In this paper, the authors prepared porous MgAlOx, but they did not mention whether the catalytic activity of Pd/M1A2 was superior to the non-porous Pd/MgAl2O4 or not. The reaction pathway changed by adding Mg component which was different from that of Pd/Al2O3. It is also known by previous papers, but how the change of the reaction pathway affected the reaction rate was not described. In the current paper, I could not find the novelty or advantage of the Pd/M1A2 for the dehydrogenation reaction. Other comments are listed below.

1)     In the introduction, the authors noted that strong interaction of PdO with MgAl2O4 suppressed the sintering of PdO and that positively charged Pd on MgAl2O4 promoted rapid desorption of products. I expected that such effect was also observed or more pronounced on aerogel MgAlOx in this work but none of the electronic properties of Pd was explained.

2)     Size distribution, average Pd particle size, and deviation of Pd size based on TEM should be shown. The magnification of Fig. 3d and f was too low.

3)     Single electron transfer using EPR is directly related to acidity and/or basicity of supports?

4)     In the section 2.1, the authors stated that M4A1 was expected to be the most promising support for Pd dehydrogenation catalysts in terms of selectivity. However, Pd/M1A2 showed the highest activity with excellent selectivity. Pd/M2A1, which showed 100% selectivity, was even higher than Pd/M4A1. The authors should explain the incompatibility.

5)     The authors stated that basic sites related to Mg can relieve some of the sterical hindrance opening up a more thermodynamically favorable pathway. However, no experimental evidence was given.

Author Response

The authors of the manuscript express their gratitude to the reviewer for a careful reading of the text. Reply to Reviewer 2 submitted as a doc file

Reviewer 3 Report

The authors prepared a series of catalysts of Pd Supported on Aerogel-Prepared Mg-Al Oxides for N-Heterocycle Dehydrogenation. The effects of Mg:Al ratio on catalyst property and the catalytic activity are well investigated. The results are interesting and my questions are as follows.

1.       Line 50, it is said that heterocyclic molecules have lower dehydrogenation temperatures. What is the dehydrogenation temperature? The author should give a temperature value or a temperature range.

2.       Line 166, the increase of the aluminum content makes it possible to produce materials with higher surface area. Then, why do M1A2 and M1A4 have a higher surface area than the pure Al2O3?

3.       Line 184, the average dimension for MgO is 4-5nm. What kind of method the author use to calculate the MgO dimension?

4.       Line 187, the mixed oxide contains cationic vacancies to compensate for the extra charge of Al3+ cations. Some more detail information should be given to support this point.

Author Response

The authors of the manuscript express their gratitude to the reviewer for a careful reading of the text. Reply to Reviewer 3 submitted as a doc file

Round 2

Reviewer 2 Report

This manuscript has been revised from the previous reports, but it still needs major revision. Please see the following comments:

1)     In authors’ reply 1, they stated that the electronic properties of Pd was beyond the scope of this study. However, the electronic properties of Pd change depending on the size of Pd particles and greatly affect the catalytic activity. Therefore, the origin of the high catalytic performance of Pd/MgAl2O4 cannot be discussed without the electronic state of Pd.

2)     In authors’ reply 2, my comment, “low magnification of Fig. 3d and f” means that please provide the image taken like Fig. 3b to demonstrate clearly whether Pd clusters (<1 nm) were present or not. If Pd clusters were present on only MgO, the electronic state and redox properties of Pd greatly alters from the Pd on other metal oxide supports. In p. 7, the authors discussed the size of Pd nanoparticles and ignored the Pd clusters (<1 nm). Although the authors showed the CO chemisorption results, the Pd particle size estimated by CO chemisorption.

3)     In authors’ reply 5, they stated that their previous work on computational results of the conformation of the substrate molecule revealed that dehydrogenation of the central ring is hindered due to steric hindrance of the substrate molecule on Pd. The previous work did not mention how Al2O3 support contribute to the adsorption of the substrate and occurrence of the steric hindrance did not given. Either, the reason why steric hindrance was resolved by the addition of Mg (or basic sites) was not given. The explanation “basic sites related to Mg can relieve some of the sterical hindrance” is just speculation based on the observation of catalytic reactions, not the experimental evidence that show how the presence of Mg affect the adsorption of the substrate to resolve the steric hindrance.

4)     Figure 6 is not “reaction mechanism”, just show “reaction pathway”. How basic sites on MgAlOx participate in the dehydrogenation should be described in the “reaction mechanism”.

Author Response

Reply to reviewer 2

Thank you very much for your valuable comments. Please see our replies to your specific comments below.

  1. In authors’ reply 1, they stated that the electronic properties of Pd was beyond the scope of this study. However, the electronic properties of Pd change depending on the size of Pd particles and greatly affect the catalytic activity. Therefore, the origin of the high catalytic performance of Pd/MgAl2O4 cannot be discussed without the electronic state of Pd.

Reply

We totally agree with the reviewer that the electronic properties of Pd species are important. Still, we have repeat again that it was beyond the scope of this study. Its goal was to synthesize a series of mixed Mg-Al oxides by the aerogel method, study their properties, prepare Pd catalysts supported on them and study the effect of the support on their catalytic activity, demonstrating that this approach is promising for preparation of a larger batch of such catalysts and their study in more detail in the future. In particular, we believe that in this case it is preferable to synthesize catalysts with several different Pd concentrations only over the most interesting supports.

In any case, XPS study was not performed when the experiments described in the manuscript were performed. Synthesis of another batch of catalysts and their detailed characterization, including performing an XPS study, will take much more time than we have for the preparation of a revised version of the manuscript.

  1. In authors’ reply 2, my comment, “low magnification of Fig. 3d and f” means that please provide the image taken like Fig. 3b to demonstrate clearly whether Pd clusters (<1 nm) were present or not. If Pd clusters were present on only MgO, the electronic state and redox properties of Pd greatly alters from the Pd on other metal oxide supports. In p. 7, the authors discussed the size of Pd nanoparticles and ignored the Pd clusters (<1 nm). Although the authors showed the CO chemisorption results, the Pd particle size estimated by CO chemisorption.

Reply

We have modified Figure 3 and added TEM images with 10 nm scale bar having higher magnification. Due to the structural lability of the aerogel Pd/Al2O3 and Pd/Mg-Al-Ox catalysts under the electron beam, images with very high magnification are difficult to focus. The smallest Pd nanoparticle observed in these images are about 2 nm in size. So, no Pd nanoparticles smaller than 1 nm are not observed by TEM.

The analysis of the literature data suggests that small Pd clusters are likely to be present in supported Pd catalysts. For example, such clusters in Pd/Al2O3 were earlier demonstrated to have higher specific catalytic activity in CO oxidation than larger Pd nanoparticles [48]. Still, we do not observe such species by TEM, do not know their relative abundance, and do not have any experimental data to discuss their relative activity in comparison with larger Pd nanoparticles.

  1. In authors’ reply 5, they stated that their previous work on computational results of the conformation of the substrate molecule revealed that dehydrogenation of the central ring is hindered due to steric hindrance of the substrate molecule on Pd. The previous work did not mention how Al2O3 support contribute to the adsorption of the substrate and occurrence of the steric hindrance did not given. Either, the reason why steric hindrance was resolved by the addition of Mg (or basic sites) was not given. The explanation “basic sites related to Mg can relieve some of the sterical hindrance” is just speculation based on the observation of catalytic reactions, not the experimental evidence that show how the presence of Mg affect the adsorption of the substrate to resolve the steric hindrance.

Reply

Figure 6 was amended with Figures 6b and 6c showing the proposed reaction mechanisms. A paragraph with their description was added to the text of the manuscript.  Also an additional reference 54 was added.

According to the proposed mechanism, over Pd/Al2O3 dehydrogenation occurs in one stage by abstraction of two hydrogen atoms by a Pd nanoparticle (Fig. 6b). In this case, both hydrogen atoms must be oriented in roughly the same direction towards the Pd nanoparticle, which is apparently possible only for the side ring due to the sterical problems. Over supports with Mg, strong basic sites can participate in the hydrogen abstraction (Fig. 6c). In this case, one hydrogen atom is initially abstracted by a basic site whereas the second atom is taken by a Pd nanoparticle in the following step. Thus, the sterical problems are lifted, and the dehydrogenation process can follow the most energetically favorable pathway. In both cases, eventually coupling of the hydrogen atoms yields molecular hydrogen.

  1. Figure 6 is not “reaction mechanism”, just show “reaction pathway”. How basic sites on MgAlOx participate in the dehydrogenation should be described in the “reaction mechanism”.

Reply

The authors agree with this comment. Figure 6 was restructured as it was discussed in our previous reply, and images with the proposed reaction mechanisms were added.
